# EVAR Follow-Up with Ultrasound Superb Microvascular Imaging (SMI) Compared to CEUS and CT Angiography for Detection of Type II Endoleak

**DOI:** 10.3390/diagnostics12020526

**Published:** 2022-02-18

**Authors:** Marco Curti, Filippo Piacentino, Federico Fontana, Christian Ossola, Andrea Coppola, Paolo Marra, Antonio Basile, Anna Maria Ierardi, Gianpaolo Carrafiello, Giulio Carcano, Matteo Tozzi, Gabriele Piffaretti, Massimo Venturini

**Affiliations:** 1School of Medicine and Surgery, Insubria University, 21100 Varese, Italy; federico.fontana@uninsubria.it (F.F.); christian.ossola@asst-settelaghi.it (C.O.); giulio.carcano@uninsubria.it (G.C.); matteo.tozzi@asst-settelaghi.it (M.T.); gabriele.piffaretti@asst-settelaghi.it (G.P.); massimo.venturini@uninsubria.it (M.V.); 2Diagnostic and Interventional Radiology Department, Circolo Hospital, ASST Settelaghi, 21100 Varese, Italy; filippo.piacentino@asst-settelaghi.it (F.P.); andrea.coppola@asst-settelaghi.it (A.C.); 3Department of Diagnostic Radiology, Giovanni XXIII Hospital, Milano-Bicocca University, 24127 Bergamo, Italy; pmarra@asst-pg23.it; 4Radiology Unit I, Department of Medical Surgical Sciences and Advanced Technologies “GF Ingrassia”, University Hospital “Policlinico-San Marco”, University of Catania, 95123 Catania, Italy; basile.antonello73@gmail.com; 5Diagnostic and Interventional Radiology Department, Fondazione IRCCS Cà Granda Ospedale Maggiore Policlinico, 20122 Milan, Italy; annamaria.ierardi@policlinico.mi.it (A.M.I.); gianpaolo.carrafiello@unimi.it (G.C.); 6Department of Radiology and Department of Health Sciences, Fondazione IRCCS Cà Granda Ospedale, University of Milan, 20122 Milan, Italy; 7Department of General, Emergency and Transplants Surgery, Circolo Hospital, ASST-Sette Laghi, 21100 Varese, Italy; 8Vascular Surgery Department, Circolo Hospital, ASST-Settelaghi, 21100 Varese, Italy

**Keywords:** abdominal aortic aneurysm, endovascular abdominal aortic aneurysm repair, endoleak, ultrasound, superb microvascular imaging

## Abstract

The aim of this study was to evaluate the usefulness of superb microvascular imaging (SMI) versus contrast-enhanced ultrasound (CEUS) and compared to computed tomography angiography (CTA) as a reference standard, for detection of type II endoleak during follow-up of endovascular abdominal aortic aneurysm repair (EVAR). Between April 2017 and September 2020, 122 patients underwent post-EVAR follow-up with CTA at 3 months and with ultrasound SMI and CEUS at 4 months from the EVAR procedure. Aneurysmal sac diameter and graft patency were evaluated; endoleaks were assessed and classified. Sensitivity, specificity, positive and negative predictive values, and diagnostic accuracy were calculated both for SMI and CEUS and compared to CTA. Furthermore, the percentage of agreement and Cohen’s Kappa coefficient were calculated. CTA revealed 54 type II endoleaks. Ultrasound SMI and CEUS presented the same sensitivity (91.5%), specificity (100%), positive (100%), and negative (92.8%) predictive and accuracy (95.9%) value for detecting type II endoleak. The same percentage of agreement of 94.9% was found between SMI/CEUS, and CTA with a Cohen’s Kappa coefficient of 0.89. The diagnostic accuracy of SMI is comparable with CEUS in the identification of type II endoleaks after EVAR. Since SMI is less invasive, less expensive, and less time-consuming, this method may be considered to be a potential tool for monitoring patients after EVAR implantation.

## 1. Introduction

Nowadays, the endovascular treatment of abdominal aortic [1,2,3,4,5] and visceral aneurysms [6,7,8] with endoprosthesis represents the first therapeutic choice in many hospitals, less burdened by morbidity and mortality than surgery. Endovascular abdominal aortic aneurysm repair (EVAR) is complicated by an endoleak—defined as the persistence of a vascular communication between the systemic circulation and the aneurysmal sac—in approximately 45% of cases [9].

Endoleaks are usually asymptomatic and may progress to aneurysm rupture [10]. For this reason, patients need a strict follow-up (FU). The ideal imaging modality in the evaluation of endoleaks should be economical, easily repeatable, safe, and accurate. Currently, computed tomography angiography (CTA) is the reference standard for EVAR FU, due to its availability, reproducibility, rapidity, and diagnostic reliability; however, CTA is burdened by high radiation dose related to the available technology, the administration of potentially nephrotoxic contrast agents, and relatively high costs [11,12,13]. Ultrasound-based techniques, such as color Doppler ultrasound [14] and, in particular, contrast-enhanced ultrasound (CEUS) are considered to be a valid alternative to CTA in EVAR FU; they are safe, not expensive, easily repeatable, and adequately accurate in the identification of endoleaks [9,15,16,17].

In January 2014, a new imaging technology was developed by Toshiba, called superb microvascular imaging (SMI). This technology allows the purification of the Doppler signal, eliminating noise and background artifacts, without reducing the vascular signal. These features overcome the limitations of traditional color Doppler ultrasound in detecting microvascular blood flows [18]. SMI also filters the signal originating from tissue movement, enabling selection and analysis of low-velocity blood flows. Consequently, both high-speed and low-speed flows are also well represented within small vessels with SMI technology, which provides high image resolution with an elevated frame rate (>50 fps) [18]. SMI can be displayed in two different modes, i.e., color SMI and monochrome SMI. The color SMI mode shows conventional grayscale ultrasound B-mode with superimposed color Doppler signals on the same image. Conversely, the monochromatic mode displays only vascular structure information which is amplified by eliminating background signals [18]. As compared with the traditional color Doppler ultrasound (CDUS), monochromatic SMI is more sensitive to slower blood flow, reproducing images which are similar to those obtained by CEUS, without the use of intravenous contrast media.

The aim of this work is to evaluate the diagnostic effectiveness of SMI as an alternative to CEUS and CTA for the detection of post-EVAR endoleak, using CTA as the reference method.

## 2. Materials and Methods

In the period between September 2017 and September 2020, 122 patients treated with EVAR who underwent abdominal CTA scan for the 3-month follow-up were enrolled in this study.

The EVARs were all performed at a single institution; preoperative and intraoperative planning was made in collaboration with the department of vascular surgery.

In 96 (85%) patients, an infrarenal fixation was performed while suprarenal fixation was chosen in 16 patients (15%).

CTA was performed on an outpatient basis using a 64-layer multislice scanner (Aquilon 64, Toshiba, Canon Medical, Rome, Italy) with the following parameters: collimation 64 × 0.5 mm, helical mode, tube voltage 120 kV, rotation time 0.5, and pitch 0.8. The three-phase scanning protocol included one precontrast scan, an angio phase scan (using the bolus tracking technique), and a venous phase scan about 100 s after contrast media administration. The SMI and CEUS examinations were performed with a ultrasound scanner Toshiba’s Aplio^TM^ 500 (Canon Medical, Rome, Italy) equipped with a 1–6 MHz convex ultrasound transducer by two experienced radiologists, who were blinded to the results of the CTA. Firstly, one radiologist (R1) started the ultrasound examination in B-mode for a morphological study of the abdominal aorta, from the diaphragm to the iliac arteries, and for assessment of the maximum axial diameters of the aneurysm sac. Subsequently, the aneurysmal sac was evaluated with SMI technology. The presence of endoleak was defined as the evidence of hyperechoic foci within the sac, which was examined on axial and sagittal scans on a side-by-side monitor preset.

Then, a second radiologist (R2), after the aforementioned B-mode scan, performed a CEUS, after the intravenous administration of a 5 milliliter bolus of contrast agent (SonoVue, Bracco, Milano, Italy). The following parameters were assessed: diameter of the aneurysm, patency of the endoprosthesis, as well as identification and classification of the endoleak, if present. The maximum antero-posterior aneurysm sac diameter was considered [19]. All the ultrasound examinations were timed, with separate counting of the three scan phases, B mode, SMI, and CEUS. The images were analyzed using conventional post-processing techniques.

All procedures performed in this study were in accordance with the ethical standards of the institutional and national research committee and with the 1964 Declaration of Helsinki and its later amendments or comparable ethical standards. Approval for this specific study was obtained by the institutional review board, according to the National Policy in the matter of Privacy Act on retrospective analysis of anonymized data; informed consent, as stated by Legge 22 Dicembre 2017 n.219 Gazzetta Ufficiale della Repubblica Italiana, was signed by each patient.

### 2.1. Outcome Measures

The following data were collected: age, sex, comorbidities (hypertension, dyslipidemia, diabetes, BMI, previous cardiovascular surgery, chronic kidney disease(CKD)), aneurysm size, time between EVAR placement and CTA, time between EVAR placement and SMI and CEUS, patency of the endoprosthesis, presence of endoleaks detected with CTA, SMI or CEUS, classification of endoleak, ultrasound feasibility for adequate acoustic window, and ultrasound scan time.

### 2.2. Statistical Analysis

The diagnostic accuracy of endoleak detection with SMI techniques and CEUS was evaluated in terms of sensitivity, specificity, positive predictive value (PPV), negative predictive value (NPV), and accuracy as compared with CTA, considered the reference standard. The percentage of agreement and Cohen’s Kappa coefficient were calculated to compare the different methods.

## 3. Results

The sample of 122 patients enrolled included 110 males (mean age = 76.7 years) and 12 females (mean age = 76.8 years) who had undergone EVAR procedures. Patients’ comorbidities are listed in Table 1.

A flowchart of patients, included and excluded, and endoleak detection results are listed in Figure 1.

The mean diameter of the aneurysm sac was 59.2 mm (range 48.1–77.4 mm). The median period of time between the EVAR procedure and CTA was 92 days (interquartile range 82–100). The median period of time between the CTA and ultrasound examination was 31 days (interquartile range 28–35). All the endoprosthesis were proven to be patent both at CTA and ultrasound examinations (Figure 2).

CTA scan detected 57 endoleaks (57/122, 46.7%): 54 were type II (39 through lumbar arteries and 15 through the inferior mesenteric artery) (Figure 3 and Figure 4), 2 were type Ia, and and 1 was type III.

Patients with type I and III endoleaks were treated in emergency and excluded from the study. Therefore, 119 of 122 patients were included in the study and underwent the subsequent ultrasound control (Table 2).

In 49 patients (49/54, 90.7%) SMI and CEUS agreed with CTA on the presence of type II endoleak.

In 5 cases (5/119, 9.3%) neither SMI nor CEUS recognized a CTA detected type II endoleak.

In no cases, did SMI and CEUS detect any endoleak in patients with a negative CTA. The findings of SMI and CEUS were concordant in all patients.

In 4 cases (3.6%), it was necessary to reschedule the ultrasound study, due to the presence of extensive interposed enterocolic meteorism. The patients were instructed to fast and take gas reducing drugs, so it was possible to perform the examination.

The ultrasound scans took 3–5 min (mean 4.3 min) for B-mode assessment, 3–5 min (mean 4.1 min) for SMI, and 6–7 min (average 6.10 min) for CEUS.

In summary, sensitivity, specificity, positive and negative predictive values, and accuracy with SMI and CEUS as compared with CTA were 91.5%, 100%, 100%, 92.9%, and 95.9%, as shown in Table 3.

The percentage of agreement between SMI and CTA was 95.8% with a Cohen’s Kappa coefficient of 0.915.

## 4. Discussion

Endovascular treatment of abdominal aorta aneurysm (EVAR) is a recognized alternative to open surgery [20]. EVAR is burdened by a lower mortality and complication rate [21]; however, it requires a “life-long” FU in which imaging plays a key role to identify complications such as fractures or stent migration, thrombosis, infections, sac enlargement, and endoleaks [22]. An endoleak is defined as reperfusion of the aneurysmal sac and represents the “Achilles’ heel” of the procedure, being the most frequent complication after EVAR procedure [23,24,25], with a reported incidence around 45% [9]. Endoleaks are classified in five types [9]. In type I and III endoleaks, prompt reoperation is recommended because they are high-flow alterations with an elevated risk of rupture. Type II endoleaks are the most frequent, due to a retrograde blood flow to the sac coming from collateral branches (e.g., lumbar or inferior mesenteric artery); in these specific cases, FU is recommended to monitor possible sac size increase [23]. According to the literature, the reference standard for post-EVAR FU is CTA, thanks to its availability, rapidity, anatomical panoramic view and uniformity of protocols; CTA can help to exactly estimate the aneurysmal sac diameters and to accurately detect complications [25,26]. However, some drawbacks limit the use of CTA: high ionizing radiation dose (>20 mSv for a thoraco-abdominal CT scan), nephrotoxicity related to iodinated contrast medium administration, static view with inability to identify the direction of blood flow, and finally, high costs [26]. Therefore, over the years, there have been efforts to search for a safer and more economic method as an alternative to CTA for post-EVAR FU [24].

An additional technique is magnetic resonance (MR). MR offers the advantages of no radiation exposure, less or no renal toxicity of contrast agents, and information about flow direction. Nevertheless, it has three main setbacks: classic MRI contraindications, magnetic susceptibility artifacts, and radiofrequency shielding effects [23,27]. Studies that have compared MR with CT imaging have demonstrated that MRI had a higher sensitivity to detect type II endoleaks. MRI with contrast medium and late acquisitions (3–30 min after the injection) could display endoleaks undetectable by CT [28]. Finally, recent studies have reported interesting data on the diagnostic value of a non-contrast MRI in the detection of endoleak which allow for spare contrast agents [29]. However, due to the non-extensive availability of this method, its high cost, and the long duration, it is not frequently included in the post-EVAR follow-up. An MRI should be considered in patients with a continuous growth of the aneurysmal sac and negative or unclear findings at CTA [28,30].

Ultrasound surveillance is an inexpensive, non-invasive method that does not use ionizing radiation and allows for evaluate the diameters of the aneurysmal sac, with an accuracy comparable to CT [25]. Despite this, it is an operator-dependent tool and might underestimate the diameter of the sac as compared with CTA. Regarding the identification of endoleaks, data reported in the literature claim that color Doppler ultrasound (CDUS) is characterized by high specificity (94%) and relatively low sensitivity (77%), with a notable frequency of false positive and false negatives; these limits affect the use of CDUS in the surveillance after EVAR [31].

After the introduction of the 2nd generation ultrasound contrast medium which are non-nephrotoxic and well tolerated by patients, and thanks to the development of dedicated software, the use of CEUS in post-EVAR FU has markedly increased in the last years [32]. Recent publications have demonstrated that CEUS is a better technique to evaluate endoleaks as compared with CDUS, thanks to its ability to provide real-time information about the direction and velocity of blood flow; this allows for identification of late low-flow endoleaks, which are hardly visible with CTA [31,33,34,35,36]. In 2012, Karthikesalingam et al. reported sensitivity values of CEUS in the identification of the endoleaks between 90–97% [35]. In addition, Mirza et al. showed high sensitivity values (94%), with 88% specificity [31]. The high values for endoleak detection with CEUS were due to its high sensitivity to identify low-flow endoleaks as compared with CTA [30]. Moreover, CEUS, thanks to its ability to visualize in real time the direction and type of blood flow, provides endoleak characterization [37]. However, CEUS is burdened by the need of a highly experienced operator and expensive contrast agents.

In 2014, an algorithm called “superb microvascular imaging” was introduced by Toshiba; SMI generates images similar to those obtained with CEUS without intravenous contrast injection [32]. Few authors have analyzed the sensitivity values, specificity, and accuracy of SMI versus CEUS and CTA for endoleaks detection [32,36]. Cantisani et al., in a cohort of eight type-II endoleak patients, found that SMI was less sensitive than CEUS and CTA, but it was reliable in the classification of endoleaks. Thus, he proposed the use of SMI in post-EVAR FU especially when CEUS (acute coronary syndrome or unstable angina) or CTA (renal failure) were contraindicated [32]. Gabriel et al. analyzed a larger group of patients with a higher number of endoleaks (*n* = 15, three patients with type Ia, nine patients with type II, and three patients with type III) examined by SMI, CEUS, and CTA and he reported equivalence among the three methods for endoleak evaluation, with comparable sensitivity, specificity, and accuracy values (100%, 93%, and 97%, respectively) [36].

We found a higher level of agreement between SMI and CEUS; all the endoleaks identified by SMI were confirmed by the second radiologist who performed the subsequent CEUS scan after injection of contrast medium. Furthermore, the comparison with CTA showed that SMI was characterized by high sensitivity (91.5%) and specificity (100%) for endoleak identification. In only five cases, SMI failed to identify the presence of endoleak, which had been visualized by CTA. We observed no false positives.

Based on these findings, SMI could be a suitable method to control the evolution of type II endoleaks [32]. The limits of the SMI algorithm are the same as for CEUS and general ultrasound: it is hindered by intestinal gas, high body mass index, and eventual post-surgical subcutaneous emphysema. Finally, SMI does not give a quantitative assessment but only a qualitative one; indeed, it can only identify the presence of an endoleak but does not provide information about flow direction. However, this limitation is not very relevant because once the endoleak is identified, these patients could continue routine FU on an annual basis with SMI if the diameter of the sac is stable; otherwise, a CTA scan is recommended. Another important limitation of SMI is its current availability on a single ultrasound platform.

In conclusion, SMI can be proposed as a valid and less invasive alternative technique to CTA and CEUS for endoleak detection after EVAR; however, further studies are necessary to confirm its reliability.

## Figures and Tables

**Figure 1 diagnostics-12-00526-f001:**
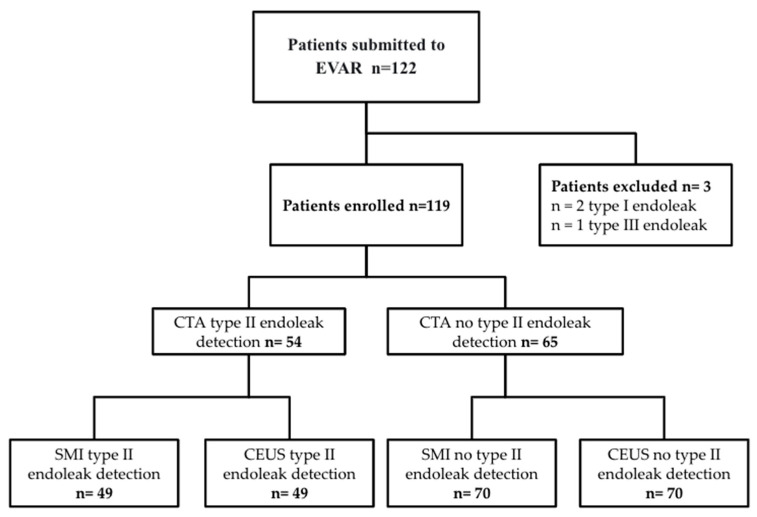
Flowchart of all EVAR (2017–2020) included in the study and endoleaks detected with SMI, CEUS, and CTA.

**Figure 2 diagnostics-12-00526-f002:**
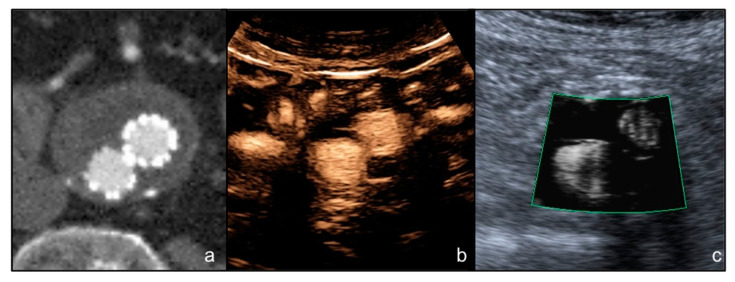
(**a**) A male patient aged 80 years. Arterial phase axial CT image shows the presence of EVAR with no endoleak and patency of the prosthetic iliac branched; (**b**) axial CEUS image of the distal portion of the EVAR, shows no sign of endoleak and patency of the endoprosthesis iliac branches; (**c**) ultrasound image acquired at the same level as (**b**) with SMI mode clearly shows the patency of the prosthetic branches without endoleaks.

**Figure 3 diagnostics-12-00526-f003:**
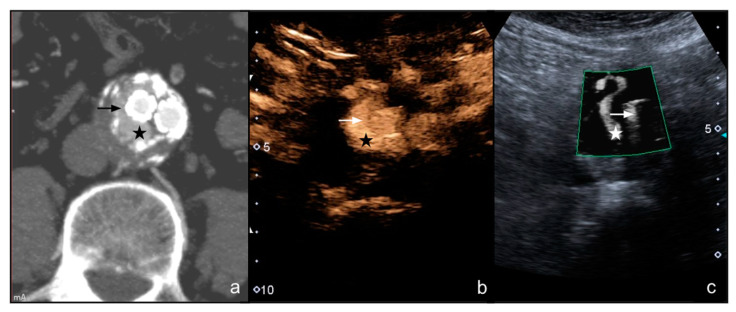
(**a**) A male patient aged 67 years. Arterial phase axial CT image shows the presence of EVAR with type II endoleak (star) surrounding the right iliac branch (arrow); (**b**) axial CEUS image of the distal tract of the EVAR shows the type II endoleak (star) surrounding the right iliac prosthetic branch (arrow), with a typical tubular shape; (**c**) SMI image on the same plane as (**b**) clearly shows the presence of type II endoleak (star) with a comparable tubular shape.

**Figure 4 diagnostics-12-00526-f004:**
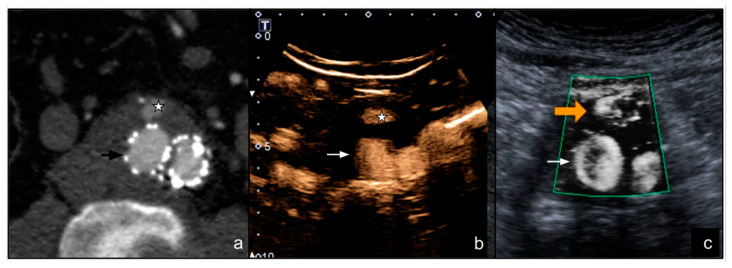
(**a**) A male patient aged 78 years. Arterial phase axial CT image shows the presence of EVAR with a type II endoleak (star) located anteriorly to the right prosthetic branch (arrow); (**b**) axial CEUS image of the distal portion of the EVAR shows the type II endoleak (star) in the anterior portion of the aneurysmal sac (arrow); (**c**) SMI image on the same plane as (**b**) clearly shows the type II endoleak (orange arrow).

**Table 1 diagnostics-12-00526-t001:** Patients’ characteristics: demographics and comorbidities of the entire cohort (*n* = 119).

**Variable**	
Demographic data	
M:F	110:12
Age (years, mean)	Males 76.72 years Females 76.83 years
**Comorbidities**	
Hypertension	95 (84.82%)
Dyslipidemia	83 (74.10%)
Diabetes	60 (53.57%)
BMI (average)	28.9
Previous CV surgery	17 (15.17%) 12 Carotid 5 valve surgery
CKD, (eGFR<30 mL/min)	28 (25%)

**Table 2 diagnostics-12-00526-t002:** Results of endoleak detection with SMI, CEUS, and CTA.

	No Endoleak Detection	Endoleak Detection
SMI	70	49
CEUS	70	49
CTA	65	54

**Table 3 diagnostics-12-00526-t003:** Statistical results of sensitivity, specificity, positive and negative predictive values, and accuracy of SMI and CEUS examinations as compared with CTA.

	Sensitivity	Specificity	Positive Predictive Value	Negative Predictive Value	Accuracy
SMI	91.53%	100.00%	100.00%	92.86%	95.97%
CEUS	91.53%	100.00%	100.00%	92.86%	95.97%

## Data Availability

The data presented in this study are available on request from the corresponding author.

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
