# Peer review of "EVAR Follow-Up with Ultrasound Superb Microvascular Imaging (SMI) Compared to CEUS and CT Angiography for Detection of Type II Endoleak"

_diagnostics, 2022, doi:10.3390/diagnostics12020526_

Round 1

Reviewer 1 Report

“Endovascular aneurysm repair (EVAR) follow-up with ultrasound with superb microvascular imaging (SMI) compared to contrast enhanced ultrasound (CEUS) and CT angiography in detecting type II endoleak” is a multicenter prospective cohort study from Italy.

The study enrolled patients from 2017 to 2020. The aim was to evaluate the usefulness of Superb Micro-vascular Imaging as an alternative to Contrast-Enhanced Ultrasound using computed tomography angiography as gold standard for detection of type II endoleak in patients followed up after endovascular abdominal aortic aneurysm repair (EVAR).

I would suggest improving the manuscript with the flowchart of the included patients and examinations. Moreover there are no tables regarding the results described and I have found some discrepancies between results and discussion. In the results, you state that 119 patients (from 122) were enrolled and diagnosed with endoleak type II, however, only 54 are displayed when compared to CTA.

Results and tables should be improved.

Author Response

As suggested we have included a flowchart about patient included or excluded and endoleak detection. In the revised manuscript, we better specified that 122 were the total patients submitted  to CTA post EVAR, with detection of 54 type 2 endoleaks and 3 other type endoleaks. Type 2 endoleaks were included in the study while other type endoleaks were excluded.

Reviewer 2 Report

This is an interesting  and potentially valuable study, yet written in poor English. Some parts of the text are difficult to understand. The manuscript needs extensive proofreading and then can be resubmitted.

Author Response

As request a language review was performed by a native with specifical technical competence.

Reviewer 3 Report

Design: prospective or retrospective study? Please add some information regarding IRB approval or ethical commission approval.

The purpose of this paper is to evaluate the usefulness of Superb Micro-vascular Imaging as an alternative to Contrast-Enhanced Ultrasound using CTA as reference standard for detection of type II endoleak in patients treated with EVAR.

Comments as follows:

Abstract

  1. Please use the term “reference standard” instead of gold standard.

Methods

  1. It is not clear why type 1 endoleaks were excluded from the study. I think that the major clinical issue is to depict type 1 endoleak in regard to the clinical indication to do some other treatment. I

Discussion

  1. Please add a paragraph discussing the potential role of contrast and non contrast MRI in detection of endoleaks.

Author Response

1) It was a prospective study.

2)in the text, on page 3, lines 114 to 120, you will find the informed consent obtained from all patients and the compliance with our Institutional review Board. "All procedures performed in this study were in accordance with the ethical standards of the institutional and national research committee with the 1964 Declaration of Helsinki and its later amendments or comparable ethical standards. Approval for this specific study was obtained by our institutional review board, according to the National Policy in the matter of Privacy Act on retrospective analysis of anonymized data; informed consent, as stated by Legge 22 Dicembre 2017 n.219 Gazzetta Ufficiale della Repubblica Italiana, was signed by each patient. "

The use of the Smi software is routinely used in micro vascularisation assessment in many abdominal diseases: this software does not involve the use of any contrast, has no biological cost as it does not use ionizing radiation; it is in fact comparable to the application of Colour in ultrasound studies.

The patients enrolled in this study did not undergo any modification of their follow-up programme, as the study with Smi was carried out at the same time as the CEUS study, which is part of our protocol follow up for EVAR. 

3) As requested we have changed “gold standard” into “reference standard”.

4) As described, type I endoleak was excluded from the study as a medical emergency requiring corrective action between the first follow-up CTA and the subsequent follow-up with SMI and CEUS, for that reason was excluded from the study.

5) As requested we have included a minor chapter discussing MRI in post EVAR follow up and its diagnostic potential regarding Endoleak detection as follows :“An additional technique is magnetic resonance (MR). MR offers the advantages of no  radiation exposure, less or no renal toxicity of contrast agents and information about flow direction. Nevertheless, it has 3 main setbacks: classic MRI contraindications; magnetic susceptibility artifacts and radiofrequency shielding effect. Studies that compared MR with CT imaging demonstrated that MRI has a higher sensitivity to detect type-2 endoleaks. MRI with contrast medium and late acquisitions (3-30 minutes after the injection), could display endoleaks undetectable by CT.  Most protocols begin with a non-contrast examination, followed by contrast-enhanced acquisitions after intravenous administration of gadolinium [28]. Contrast agent administration enhance the visualization of endoleaks. Furthermore, 3D MR angiography allows to analyze the blood kinetics such as direction of flows in the endoleak. Finally, recent studies have reported interesting data on the diagnostic value of non-contrast MRI in the detection of endoleak which allow to spare contrast agents [29].However, due to the non-extensive availability of this method, its high cost amd the long duration it is not frequently included in the post-EVAR follow-up. Moreover, the susceptibility artifacts caused by not compatible MRI stents, especially those with stainless steel components, may adversely affect image quality and interpretation, simulating stenosis or occlusion of the graft. MRI should be considered for patients with a continuous growth of the aneurysmal sac and negative or unclear findings at CTA. “

Round 2

Reviewer 1 Report

In my opinion, the present manuscript cannot be considered for publication in the current form. First, I would not include so many abbreviations in the title. Moreover, the aim is not phrased clearly, and I believe that results should be implemented with tables instead of figures alone. The discussion is very long and hard to follow for the reader, I would suggest shortening it and describing briefly the findings according to the available literature.

Author Response

As requested we have modified the title removing unnecessary abbrevations. However, we have directly included the abbreviations EVAR and CEUS, acronyms that are now widely used, as evidenced by the numerous works that bear these acronyms directly in their titles. Here are some examples:

- Liu S, Cai W, Luo Y, et al. CEUS Versus MRI in Evaluation of the Effect of Microwave Ablation of Breast Cancer [published online ahead of print, 2022 Jan 18]. Ultrasound Med Biol. 2022;S0301-5629(21)00491-9. doi:10.1016/j.ultrasmedbio.2021.11.012

 - Hahl T, Kurumaa T, Uurto I, Protto S, Väärämäki S, Suominen V. The effect of suprarenal graft fixation during EVAR on short- and long-term renal function [published online ahead of print, 2022 Jan 21]. J Vasc Surg. 2022;S0741-5214(22)00105-7. doi:10.1016/j.jvs.2021.12.081

Furthermore, we have decided to leave Superb Microvascular Imaging and its acronym in its entirety, as it is a more recent method and not so widely known.

As requested we have rephrased the aim and furthermore we have added two additional table in the results section(table 3 and table 4).

In conclusion, the discussion was shorter in the first submission, but it was requested to add a chapter regarding MRI in evar FU. However, we have removed some phrase in order to obtain a discussion easier to read.

Reviewer 2 Report

At its present form (correctons over corrections, in different colts, etc.) the manuscript is very difficult to read.

The Authors should submit new version of the paper - not the old one with billions of corrections

Author Response

Probably due to an error you have read the version with different colours which I had named "tracked changes" to simplify your task of reviewing the work. I had also submitted a "clean manuscript" version in which the colour of the text was homogeneous.

As for the many changes to the manuscript, these are the work of the native English speaker who, as your previous request, revised the work.

Further changes were made to fulfil the requests of the other two reviewers.

Reviewer 3 Report

The revised manuscript reads better and the authors have addressed precisely all my comments. even if a point-by-point response is not provided.

Author Response

Thank you for your comments.

Round 3

Reviewer 1 Report

I have no furthers comments and I will support the Editor’s final decision.

Author Response

Thank you for your comments and support.

Reviewer 2 Report

  1. Several words improperly used (wrong translation from Italian?):
  • eventual – line 37 and 234
  • velocity – line 65 and 235
  • assume – line 208
  • iodized – line 239
  1. Instead of “allow to do something” the construction ‘ allow for doing something” should be used – several places within the text
  2. “in patients” instead of “for patients” – line 253
  3. Improper number of citation – line 288

Author Response

As suggested, we have edited and corrected translation errors. We have also fixed the incorrect citation number. Thank you for your comments and support.